# The Heating Under Micro Variable Pressure (HUMVP) Process to Decrease the Level of Saponin in Quinoa: Evidence of the Antioxidation and the Inhibitory Activity of α-Amylase and α-Glucosidase

**DOI:** 10.3390/foods13223602

**Published:** 2024-11-11

**Authors:** Ligen Wu, Anna Wang

**Affiliations:** 1College of Food Science and Engineering, Henan University of Technology, Zhengzhou 450002, China; ligen2016@126.com; 2National Engineering Research Center of Wheat and Corn Further Processing, Henan University of Technology, Zhengzhou 450002, China

**Keywords:** quinoa, saponin, heating under micro variable pressure, inhibitory activity

## Abstract

To reduce the level of saponin while preserving essential nutrients and antioxidative properties in quinoa (*Chenopodium quinoa*), this study delves into the optimization of the HUMVP process and thoroughly examines its effects on antioxidation as well as its inhibitory influence on α-amylase and α-glucosidase. The optimal HUMVP conditions involved wetting quinoa grains with 6% water (pH = 6.0) and subjecting them to a 4 min treatment under 0.35 MPa pressure. The values of ^•^OH, DPPH, and ABTS^•+^ scavenging rate of the extracts from the quinoa sample (named Q2HUMVP) treated under the optimum HUMVP process were 70.02, 87.13, and 50.95%, respectively. Furthermore, the treatment preserved 95.20% of polyphenols and 73.06% of flavonoids, while the saponin content was reduced to 23.13% of that in raw quinoa. Notably, Q2HUMVP extracts demonstrated superior inhibitory activity against α-amylase and α-glucosidase compared to dehulled quinoa samples. The inhibition exhibited by the quinoa sample extracts on α-amylase and α-glucosidase was found to be reversible.

## 1. Introduction

Quinoa (*Chenopodium quinoa*), classified as a pseudocereal, is renowned for its exceptional nutritional characteristics [1,2]. Notably, it is rich in phenolic compounds, flavonoids, saponins, anthocyanins, contributing to its robust antioxidant properties in quinoa-based foods [3]. Additionally, research indicates that a diet incorporating whole-grain quinoa might reduce the lipid profiles and glucose levels in rats [4]. Moreover, the consumption of quinoa products has been shown to notably lower blood sugar and triglyceride level [5]. Interest in quinoa is growing around the world [6].

Quinoa grains can be cooked similarly to rice and are versatile ingredients used in various culinary applications, including porridge, baked goods, and extrusion products [7]. However, it is well established that the nutrient composition, bioactive compounds, and antioxidant capacity of foods can vary significantly due to different processing methods. Processing inevitably leads to alterations in nutritional properties, often resulting in the reduction of certain constituents in the food product [8]. The saponins in quinoa limit its application because of the strong bitter taste associated with them. Consequently, there have been two traditional techniques for saponin removal: dry and wet methods. Dry processing involves scouring quinoa grain under dry conditions, while wet processing entails washing the quinoa grain after scrubbing [8]. Some studies have explored the use of various types of milling equipment, such as roller, pin, or abrasive millers, to decrease the saponin level in quinoa. Research results showed that different dehulling methods yield varying production rates and levels of saponin in quinoa samples, and the nutrient composition differs among the final products [9]. Dry scouring resulted in an uneven outer surface and significant damage to the embryo, leading to a high degree of kernel fragmentation in the final products. Attempting to remove all or most of the saponins from quinoa using the dry method can result in substantial resource wastage. According to previous research findings, dry scouring quinoa grains can lead to the loss of over 11% of protein, more than 7% of fat, more than 4% of starch, more than 28% of total dietary fiber, more than 45% of soluble dietary fiber, more than 48% of saponins, more than 26% of flavonoids, and more than 42% of the total phenolics in quinoa grain. Remarkably, these significant losses occur even when only 8.6% of the grain is removed during the scouring process [10].

Dietary factors undoubtedly play an important role in the development of type 2 diabetes (T2DM) [11], by regulating and slowing down the activity of major digestive enzymes, especially pancreatic alpha amylase and alpha glucosidase, the absorption rate of carbohydrates in the gastrointestinal tract is significantly reduced, effectively controlling T2DM and alleviating symptoms of hyperglycemia [12]. The amylase and α-glucosidase in the pancreas, like delicate molecular scissors, are responsible for separating complex carbohydrates into easily absorbable disaccharides and monosaccharides, which then enter the bloodstream through the small intestine wall. Therefore, if the “scissors” can be effectively used, it will open up an effective pathway for relief, and thus, benefit T2DM patients. This strategy not only aims at the links accurately, but also shows the great potential of diet regulation in diabetes management [13].

Therefore, it becomes of paramount importance to identify a suitable method for saponin removal from quinoa while preserving its valuable nutrients, antioxidants, and functional components. In light of this, this study delves into the process of HUMVP as a means to decrease the level of saponin and retain more nutritious, antioxidative, and functional components in quinoa. This approach holds promise for facilitating the production of nutritionally rich and functional quinoa products.

## 2. Materials and Methods

### 2.1. Standards and Reagents 

The raw quinoa was donated by the College of Agriculture and Animal Husbandry, Qinghai University in the Qinghai Province, Xining City, China. 

The reagents used in this study were of analytical grade including Trolox(6-Hydroxy-2,5,7,8-tetramethylchromane-2-carboxylic acid), (purity > 97%), sodium carbonate, ferric chloride, α-amylase (A7720), pepsin (P7000), DPPH (2,2-Diphenyl-1-picrylhydrazyl), bile salt, trypsin (P7545), gallic acid, oleanolic acid, rutin, ABTS (2,2′-azino-bis(3-ethylbenzothiazoline-6-sulfonic acid) (>98%), phosphate buffer, ethanol, methanol, Folin–Ciocâlteu reagent, hydrochloric acid (37%), and acarbose (98%) and were supplied by Merck (Shanghai, China).

### 2.2. Processing of Quinoa Grains

The raw quinoa (recorded as Q2QQ) was donated from Qinghai province in China, the saponin, flavonoid, and total phenolic content of the raw quinoa grain were 24.97 mg/g, 2.63 mg/g, and 343.72 mg/100 g. The quinoa grains were dehulled using an abrasive mill, and the material was sieved, and the material over the sieves with a size of 250 μm was collected and named the dehulled grain (recorded as Q2MM), with the yield of 91.4%. The saponin, flavonoid, and total phenolics content of the dehulled quinoa grain were 20.22 mg/g, 1.82 mg/g, and 297.43 mg/100 g. 

The HUMVP equipment was the new equipment invented by our research team, for which we applied for a patent in China in 2021 [14]. The HUMVP equipment consists of two same units, as shown in Figure 1, a pipe controlled by the electromagnetic high-pressure valve (3) connects one ellipsoid tank (2) to another ellipsoid tank (5). When the quinoa grains were poured in, the air was substituted by the inert gas and heated to a certain pressure value P1 (P1 = 0.2–0.45 MPa) for 3–5 min in the ellipsoid tank (2), and the air was substituted by the inert gas to a certain pressure value P2 (P2 = P1 − 0.02 MPa) in the ellipsoid tank (5) at the same time. The electromagnetic high-pressure valve (3) was then automatically opened, the pressure of both ellipsoid tank (2) and (5) were balanced instantly, and the internal air pressure of the quinoa grains was higher than that outside of the grains, at this time. As the pressure decreases slightly and instantaneously outside, the quinoa grains expanded slightly resulting in the cortex being chapped and the tissue loosened, and then the quinoa grains were cooled down quickly. After cooling, the samples were collected and recorded as Q2HUMVP.

### 2.3. Digestion of Quinoa Samples

The digestive of the quinoa samples was prepared as follows: the quinoa flour (dry basis) samples were weighed into 50 mL polypropylene copolymer tubes and mixed thoroughly with 3.5 mL of distilled water. The suspension was kept at 37 °C for 5 min and combined with 1.5 mL of pepsin–HCl solution (1.35% *w*/*w* pepsin, 0.05 M HCl, pH 2.0), and the mixture was incubated at 37 °C for 30 min on a magnetic stirrer. The pH was brought up to 6.0 by adding 3.0 mL pH 6.4 maleate buffer (0.1 M, 10 mM CaCl_2_). To initiate digestion, 2 mL of enzyme solution (0.1 M maleate buffer pH 6.0, 10 mM CaCl_2_) containing 110 units of porcine pancreas α-amylase and 33 units of amyloglucosidase was added. Digestion was performed at 37 °C with a magnetic stirrer. Aliquots of 0.5 mL were taken at selected time intervals and immediately added to 1.5 mL of a cold ethanol solution (90% *v*/*v*). The mixture was kept in an ice bath for 10 min and then centrifuged (3000× *g*, 10 min) to separate the supernatant. The collected supernatant was lyophilized as the digestive of quinoa samples (dry basis).

### 2.4. Determination of the Content of Saponin, Flavonoid, and Total Phenolics 

The content of total polyphenolics (TPC) in quinoa was determined spectrophotometrically with Folin Ciocalteau reagent (FCR), the determination method was the same as described by Kumar et al. [15]. TPC was expressed, according to the method described by Ando et al. [16], as Gallic acid equivalent mg GAE/100 g of quinoa.

The determination of the content of total flavonoids was performed according to Jia et al. [17], with a colorimetric assay, and the results were shown with catechin equivalents (CE), which were transformed from absorbance to the concentration in the standard curve prepared from the authentic catechin reported by Han et al. [18].

The determination of saponin content in quinoa was performed in light of the method reported by Han et al. [18].

### 2.5. Evaluation of Hydroxyl Radical, DPPH, and ABTS^•+^ Scavenging Rate

The evaluation of the hydroxyl radical, DPPH, and ABTS^•+^ scavenging rate were performed as the method reported by Wu et al. [10]. The calculation of the clearance rate of the hydroxyl radical, DPPH, and ABTS^•+^ scavenging rate were performed in light of the method reported by the references [18,19,20,21,22,23,24,25].

### 2.6. Inhibitory Activities Against Enzymes

#### 2.6.1. Inhibitory Activities Against α-Amylase

The method adopted to determine the inhibitory activities against α-amylase by the extracts of quinoa samples was as described by Hemalatha, Bomzan, Rao, and Sreerama [2], and the soluble starch was used as substrate, while acarbose was the positive control in the inhibition assays. Different concentrations of acarbose were prepared with a pH 6.9 phosphate buffer.

#### 2.6.2. Inhibitory Activities Against α-Glucosidase 

The inhibitory activity of the extracts of quinoa against α-glucosidase was determined by the method described byHemalatha, Bomzan, Rao, and Sreerama [2].

#### 2.6.3. Inhibition Kinetics Against α-Amylase 

The determinations of the inhibition kinetics against α-amylase by quinoa extracts were performed in light of the method by Zhao et al. [26]. A gradient of maize starch solution (1.25, 2.5, 5, 10 mg/mL) was kept under 90 °C for 15 min until gelatinized to be the substrate. The solution of α-amylase (50 μL) was mixed with 50 μL extract from the quinoa samples, and the mixture was kept at 4 °C for 15 min. The digestion started as soon as 4 mL of the substrate was poured in. The production processes to calculate the reaction rate were the methods reported by Uysal et al. [27] and Zhao et al. [26]. The constants of inhibition kinetics of quinoa extracts against α-amylase were calculated by the Dixon equation, and then those constants determined the inhibition type. 

The inhibition kinetics of quinoa extracts against α-glucosidase were studied as follows: The substrate (maltose) was configured at different concentrations of 5, 10, 15, 20, and 25 mM, and then the substrate was mixed with α-glucosidase without the inhibitor, and with 7.5 mg/mL and 15 mg/mL of quinoa extracts in phosphate buffer pH 7.2 (0.1 M) to be incubated at 37 °C, and the content of glucose in the digest was mensurated using the glucose oxidase method. The nature of inhibition was constructed from the double reciprocal plots of enzyme kinetics relying on the Lineweaver and Burk method. Depending on the Lineweaver–Burk plots (1/S vs. 1/V), Km and Vmax values were calculated [28].

### 2.7. Statistics

Statistical analyses for the results of this paper were accomplished using SPSS 18, a statistics software supplied by IBM Corporation, New York, NY, USA. The Tukey-*b* test for *p* < 0.01 was used to establish significant differences by one-way analysis of variance (ANOVA). The Original 9.0 software was used to draw the related charts. All experiments were in triplicate.

## 3. Results

### 3.1. Effect of HUMVP on the Saponin Content in Quinoa

As shown in Figure 2A, an increase in pressure (from 0.2 to 0.45 MPa) resulted in a significant reduction in quinoa saponin levels by 24.90% (*p* < 0.01), decreasing from 19.68 to 14.78 mg/g, in comparison to the sample treated at 0.1 MPa. Additionally, the duration of HUMVP treatment influenced saponin levels, with a 12.99% variation observed as treatment time increased from 2 to 6 min. Notably, saponin levels initially rose to 20.54 mg/g from the starting value of 17.37 mg/g and then declined to 16.38 mg/g due to the pH value increasing from 5 to 8.5 (Figure 2B). The saponin level in quinoa was the highest at pH 7, demonstrating an 18.25% increase compared to levels at pH 5.

The impact of pH adjustments in the water, along with variations in pressure and treatment duration during HUMVP, was assessed for its effect on total phenolic content (TPC) and total flavonoid content (TFC). Notably, both TPC (from 298.32 to 210.12 mg/100 g) and TFC (from 1.82 to 1.23 mg/g) exhibited a significant reduction with increasing pressure, reaching their peak values at a pH of 7.0 (Figure 2C). 

To determine the optimal conditions for minimizing saponin content in quinoa samples, an investigation into the HUMVP treatment was conducted, and the results were shown in Table 1. The saponin content was significantly affected by the time, pressure, pH, and the interaction between pressure and pH during HUMVP treatment (Figure 2D). Compared to raw quinoa, the quinoa sample treated under the optimal HUMVP conditions (Q2HUMVP) retained 95.20% of polyphenols, 73.06% of flavonoids, and 57.35% of saponins. In contrast, the dehulled quinoa sample (Q2MM) retained 75.21% of polyphenols, 77.63% of flavonoids, and 67.17% of saponins. Notably, the TPC value in Q2HUMVP reached its maximum at 327.22 mg GAE/100 g, with saponin content minimized to 10.65 mg/g, the lowest among all samples, while flavonoid content measured at 1.60 mg/g. 

The optimal HUMVP conditions for quinoa, aiming to minimize saponin content and enhance bioactive antioxidation, involved the following steps: The quinoa grains were moistened with water (pH 6.0, equivalent to 6% of the quinoa in weight). Subsequently, the quinoa grains were heated to 0.35, and this pressure was maintained for 4 min. Afterwards, the pressure was released gradually at a constant drop-pressure until it returned to normal, and the quinoa samples were rapidly cooled by pumping air into the settle.

The experimental results for ^•^OH, DPPH^•^, and ABTS^•+^ scavenging rate in HUMVP-treated quinoa samples are presented in Table 1. Under the optimized conditions, the scavenging rates of ^•^OH, DPPH^•^, and ABTS^•+^ were 69.89, 91.87, and 52.18% and not 70.02%, respectively. Notably, the values of ^•^OH, DPPH, and ABTS^•+^ scavenging rate were significantly affected by the treatment time and pressure, with the most substantial impact arising from the interaction between pressure and treatment time (Table 1). 

### 3.2. Enzyme Inhibitory Activities

The inhibitory IC50 values of the quinoa sample extracts on α-amylase and α-glucosidase are shown in Table 2. The IC50 α-amylase value of the quinoa sample extracts ranged from 180.14 to 190.97%, and the IC50 α-amylase value of the Q2HUMVP was significantly higher than that of the raw quinoa sample (*p* < 0.05). Conversely, the IC50 α-glucosidase value of the quinoa sample extracts ranged from 80.37 to 86.70%. The IC50 α-glucosidase value of the Q2HUMVP was significantly higher than that of the raw quinoa sample (*p* < 0.05) and lower than that of Q2MM (*p* < 0.05). Furthermore, it is worth noting that the IC50 α-glucosidase value was much lower than the IC50 α-amylase value, which was consistent with the results in previous research literature. For instance, Gao [29] reported that the IC50 α-amylase value and the IC50 α-glucosidase of green tea extract were 4020.16 μg/mL and 4.42 μg/mL separately [29]. The IC50 α-amylase value and the IC50 α-glucosidase of Salvia eriophora (*S. eriophora*) leaf extracts were 8.88 μg/mL and 2.94 μg/mL, respectively [24]. Hemalatha [2] found the IC50 α-amylase and IC50 α-glucosidase values of whole grain quinoa extract were 163.52 μg/mL and 72.36 μg/mL separately [2]. Meng [13] extracted total flavonoid myricetin and quercetin from Hovenia dulcis Thunb by ethanol and found the IC50 α-amylase and IC50 α-glucosidase values were 32.8 μg/mL, 662 μg/mL, and 770 μg/mL and 8 μg/mL, 3 μg/mL, and 32 μg/mL, respectively [13]. However, there were different results in a study by Pramod [30], who found that the IC50 α-amylase values of ethanol extracts and water extracts from the plant *Alternanthera Pungens* Kunth in India were 6.96 μg/mL and 7.54 μg/mL, respectively, and the IC50 α-glucosidase value were 76.78 μg/mL and 70.62 μg/mL, respectively. This suggests that the inhibitory effect of the extraction on α-amylase was stronger than on α-glucosidase. The extract from the plant (*Alternanthera Pungens* Kunth) contains saponins, alkaloids, steroids, triterpenes, white anthocyanins, and other compounds. The variation in the composition ratio may be the cause of the difference in the inhibitory activity against α-amylase and α-glucosidase [31]. The observed lower inhibitory activity of the extract against α-amylase in comparison to α-glucosidase may indicate a preferred strategy for effectively regulating the release of glucose from intestinal disaccharides. This strategy involves achieving moderate amylase inhibition and strong glucosidase inhibition. However, the strong α-amylase inhibitors might cause undigested sugars in the colon to subsequently abnormally ferment, causing gastrointestinal problems [28,32].

Comparing the inhibitory activities against α-amylase and α-glucosidase of the quinoa extracts and the in vitro digestive system (Table 2), it becomes evident that the IC_50 α-amylase_ and IC_50 α-glucosidase_ values of the quinoa digestive system were several times higher than those of the undigested quinoa sample extract. Herrera [32] reported that the inhibitory activities of *fenugreek* extracts and quinoa extracts against pancreatic lipase were found to be more effective than those of the in vitro digestive process. Explaining these results is complex and multifaceted because the inhibitory effect can be influenced by enzyme activity, interactions among reaction components, changes in extract components during the extraction process, etc. Some researchers have attributed such findings to the degradation of biologically active compounds during intestinal digestion. Ercan [31] observed that the activity of the in vitro digestive processes for chickpea and the saponins extracted from *Tribulus terrestris* were lower than their activity before digestion.

### 3.3. Correlation Between IC_50 α-amylase_ and IC_50 α-glucosidase_ Values and the Content of Polyphenols, Flavonoids, and Saponins in Quinoa

The results of the correlation analysis between the IC_50 α-amylase_ and IC_50 α-glucosidase_ values and the content of polyphenols, flavonoids, and saponins in quinoa are shown in Table 3. The treatment method significantly affected the content of polyphenols and flavonoids in undigested quinoa, the content of flavonoids and saponins in the digestive quinoa samples, and the IC_50 α-glucosidase_ value in the quinoa extract (*p* < 0.05). Moreover, the treatment method had an extremely significant effect on the IC_50 α-amylase_ value in quinoa sample extract, the IC_50 α-amylase_ and IC_50 α-glucosidase_ value in quinoa digestive, and the content of polyphenols in quinoa digestive (*p* < 0.01). The IC_50 α-amylase_ and IC_50 α-glucosidase_ values of the quinoa extract were extremely significantly correlated with the polyphenol content in quinoa samples (*p* < 0.01). The content of flavonoids in quinoa samples was significantly correlated with the IC_50 α-amylase_ and IC_50 α-glucosidase_ value in quinoa digestive samples (*p* < 0.05). Additionally, the content of polyphenols and flavonoids in the digestive samples demonstrated a high correlation with the IC_50 α-amylase_ and IC_50 α-glucosidase_ values in the digestive (*p* < 0.01).

### 3.4. Inhibition Kinetics 

The inhibition kinetics of quinoa extractions on starch digestion were studied to elucidate the inhibition mechanisms and patterns. The results are illustrated in Figure 3, and the rate lines (Figure 3A,C,E) went through the origin, which indicates that the inhibition type of Q2QQ, Q2MM, and Q2HUMVP against α-amylase was reversible. Reversible inhibition implies that there is no accumulation effect in vivo, suggesting that it should be safe to develop functional foods using quinoa extracts [13]. 

The inhibition mode was determined based on the Lineweaver–Burk plots (Figure 3B,D). In these plots, the inhibition kinetics curves of Q2QQ and Q2MM intersected in the first quadrant, indicating that the inhibition type of Q2QQ and Q2MM against α-amylase was competitive. Conversely, the inhibition kinetics curves of Q2HUMVP crossed in the second quadrant, suggesting that the inhibition type of Q2HUMVP against α-amylase involved a combination of competitive and non-competitive mechanisms.

The effects of the three quinoa sample extractions on initial reaction rates are depicted in Figure 4. In these plots, the rate lines (Figure 4A,C,E) went through the origin, signifying the inhibition type of Q2QQ, Q2MM, and Q2HUMVP against α-glucosidase was reversible inhibition. Lineweaver–Burk plots, presented in Figure 4B,D, illustrate that the inhibition kinetics curves of Q2QQ, Q2MM, and Q2HUMVP for α-glucosidase crossed in the X-axis, implying the inhibition type of Q2QQ, Q2MM, and Q2HUMVP on α-glucosidase was non-competitive. The results were consistent with the findings of Nagmoti and Juvekar [28], who observed that the mechanism of the *P. dulce seeds* extract against α-glucosidase inhibition was of a reversible, non-competitive nature.

## 4. Conclusions

Decreasing the saponin content in quinoa is an essential step in the production of quinoa-based foods to mitigate the bitter taste. However, this process often leads to significant nutrient and bioactive material loss. The HUMVP process applied to quinoa effectively decreased saponin levels while preserving more nutrients and bioactive functional components. Compared to raw quinoa, the optimal HUMVP treatment (Q2HUMVP) retained 95.20% of polyphenols, 73.06% of flavonoids, and 23.13% of saponins, maintaining higher levels of TPC and TFC and a lower level of saponin than dehulled quinoa. Furthermore, the Q2HUMVP quinoa extract exhibited superior inhibitory activity against α-amylase and α-glucosidase compared to dehulled quinoa samples (Q2MM). Interestingly, the inhibitory activity of the digestive system of the optimized quinoa (Q2HUMVP) against α-amylase and α-glucosidase did not significantly differ from that of raw quinoa (Q2QQ). The type of inhibition observed in the quinoa samples (Q2QQ, Q2MM, Q2HUMVP) against α-amylase and α-glucosidase was reversible inhibition. Further studies are required to gain a deeper understanding of the effects on the composition, structure, and activities of the polyphenols, flavonoids, and saponins involved in quinoa processing. In conclusion, this study’s primary goal was to assess the impact of HUMPV conditions on quinoa’s ability to retain nutrients while lowering saponins and the findings support this purpose.

## Figures and Tables

**Figure 1 foods-13-03602-f001:**
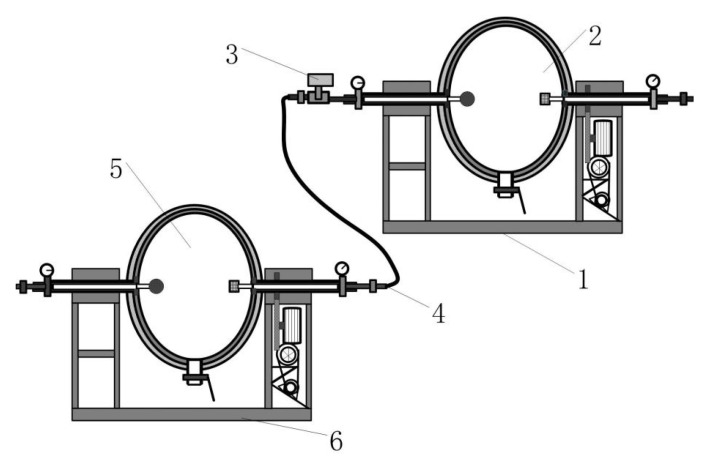
The sketch map of heating under micro variable pressure (HUMVP) equipment. 1—the HUMVP equipment unit; 2—ellipsoid tank; 3—the electromagnetic high-pressure valve; 4—pipe; 5—ellipsoid tank; 6—the HUMVP equipment unit.

**Figure 2 foods-13-03602-f002:**
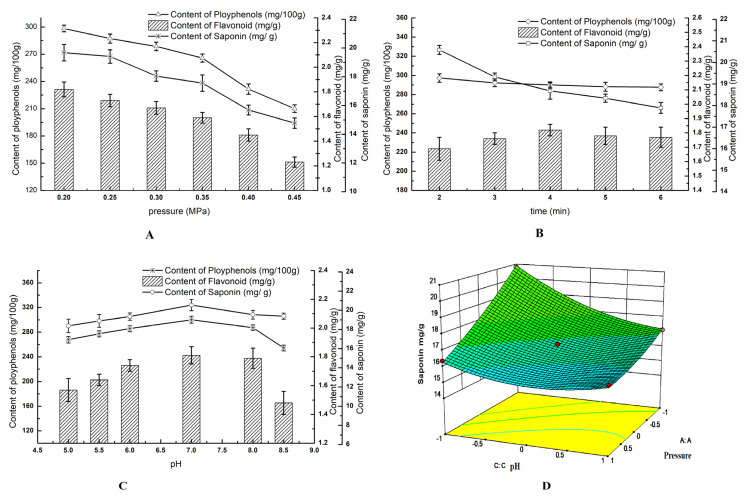
Effect of HUMVP conditions on saponin, polyphenol, and flavonoid content. (**A**)—effect of pressure on content; (**B**)—effect of time on content; (**C**)—effect of pH on content; (**D**)—effect of the interaction between pressure and pH on saponin content in quinoa.

**Figure 3 foods-13-03602-f003:**
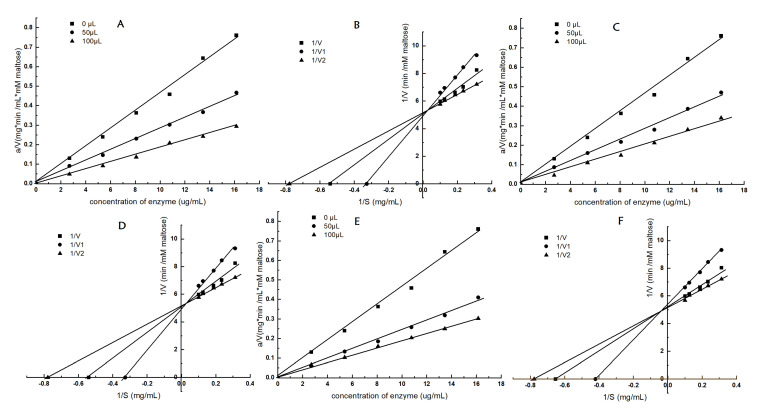
Inhibition kinetics and Lineweaver–Burk plots of α-amylase by quinoa samples: ((**A**)—Q2QQ, (**C**)—Q2MM, (**E**)—Q2HUMVP; double reciprocal plots (**B**)—Q2QQ, (**D**)—Q2MM, (**F**)—Q2HUMVP), (1/V, no inhibition; 1/V1, 50 µL inhibition; 1/V2, 100 µL inhibition).

**Figure 4 foods-13-03602-f004:**
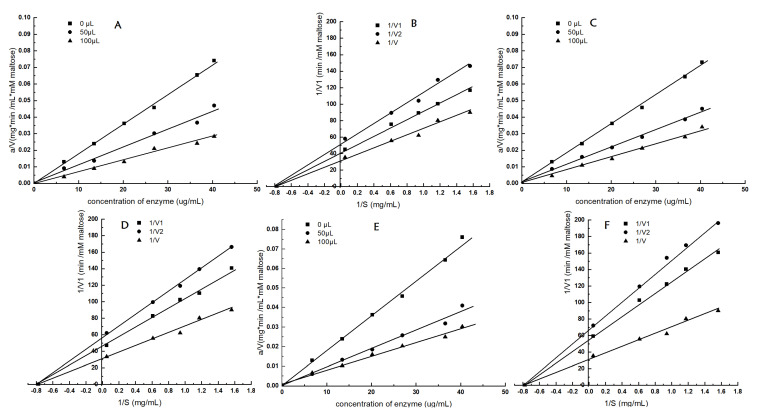
Inhibition kinetics and Lineweaver–Burk plots of α-glucosidase by quinoa samples: ((**A**)—Q2QQ, (**C**)—Q2MM, (**E**)—Q2HUMVP; double reciprocal plots (**B**)—Q2QQ, (**D**)—Q2MM, (**F**)—Q2HUMVP), (1/V, no inhibition; 1/V1, 50 µL inhibition; 1/V2, 100 µL inhibition).

**Table 1 foods-13-03602-t001:** The optimum HUMVP treatment depending on the content of saponin, TPC, flavonoid, and antioxidant activity in quinoa.

Run	Pressure /(MPa)	Time /(min)	pH	Content of Polyphenol /(mg/100 g)	Content of Flavonoid /(mg/g)	Content of Saponin /(mg/g)	^•^OH Scavenging Rate/(%)	DPPH Scavenging Rate/(%)	ABTS^•+^ Scavenging Rate/(%)
1	0.35	3	5.5	291.19	1.59	14.21	54.61	91.02	50.98
2	0.3	4	5.5	311.22	1.43	13.37	73.53	92.34	52.64
3	0.25	4	6.0	298.27	1.73	18.01	67.68	90.67	52.08
4	0.3	4	5.5	302.87	1.73	18.74	73.02	92.36	52.74
5	0.25	5	5.5	291.05	1.72	13.59	52.13	88.65	50.68
6	0.35	4	5.0	301.76	1.58	16.15	67.16	90.91	52.53
7	0.3	4	5.5	299.27	1.69	18.44	73.66	92.45	52.78
8	0.35	4	6.0	327.22	1.33	10.65	69.89	91.87	52.18
9	0.3	4	5.5	307.22	1.34	11.07	74.63	92.57	52.72
10	0.3	4	5.5	318.41	1.71	18.91	73.87	92.75	52.78
11	0.35	5	5.5	291.85	1.71	16.98	70.02	87.12	50.95
12	0.3	3	5.0	311.21	1.70	17.53	64.56	88.63	52.02
13	0.3	3	5.0	314.17	1.69	17.86	50.12	84.02	49.98
14	0.3	5	6.0	289.65	1.63	16.86	68.45	87.62	50.56
15	0.25	3	5.5	298.49	1.92	19.01	63.55	87.94	52.14
16	0.25	4	5.0	311.78	1.83	20.71	62.38	89.03	52.8
17	0.25	3	6.0	279.21	1.74	19.04	53.88	85.66	50.64
			df	TPC	TFC	Saponin	^•^OH Scavenging Rate	DPPH Scavenging Rate	ABTS^•+^ Scavenging Rate
Linear		A	1	−2.12 ***	−0.13 ***	−1.51 ***	1.37 ***	0.542 **	−0.064 *
		B	1	−3.13 ***	0.016 *	−0.63 ***	2.35 ***	−0.54 ***	0.11 **
		C	1	0.91 *	−0.054 ***	−1.05 ***	1.06 **	0.066	0.20 ***
Interactive		AB	1	1.40 **	0.077 ***	0.22 *	2.53 ***	−1.78 ***	−0.18 ***
		BC	1	10.68 ***	0.075 **	−0.072	1.66 *	−0.48 *	0.41 ***
		AC	1	−4.66	−0.028 *	0.81 ***	−0.097	−0.045	−0.94 ***
		R-Squared		0.9982	0.9921	0.9958	0.9952	0.9955	0.9953

Key to short forms: A = pressure, B = time, C = pH, AB = pressure × time, AC = pressure × pH, BC = time × pH, values shown are regression coefficients of respective terms. Figures in parenthesis denote standard error. * *p* < 0.05, ** *p* < 0.01, *** *p* < 0.001, other values are non-significant. Flavonoid = total flavonoid content, TPC = total phenolic content, Saponin = saponin content. (*N* = 3).

**Table 2 foods-13-03602-t002:** Inhibition activity of extraction from quinoa on α-amylase and α-glucosidase (µg/mL).

	Extraction of Quinoa Samples		Digestive of Quinoa Samples	
Samples	IC_50 *α*-Amylase_	IC_50 *α*-Glucosidase_	IC_50_ *_α_*_-Amylase_	IC_50_ *_α_*_-Glucosidase_
Q2QQ	180.14 ± 3.54 ^a^	80.37 ± 1.59 ^a^	901.68 ± 6.14 ^a^	399.01 ± 7.39 ^a^
Q2MM	196.97 ± 3.47 ^c^	86.70 ± 2.13 ^c^	924.21 ± 4.68 ^b^	408.97 ± 5.36 ^b^
Q2HUMVP	186.51 ± 3.89 ^b^	83.02 ± 2.97 ^b^	901.06 ± 4.39 ^a^	396.61 ± 4.31 ^a^
Acarbose	7.21 ± 0.23	83.65 ± 2.31	/	/

Note: Mean ± SD (standard deviation); means in the same column with different superscripts are significantly different (*p* < 0.01) (*N* = 3);/means no experiment.

**Table 3 foods-13-03602-t003:** The relativity of polyphenol, flavonoid, and saponin contents in quinoa samples to the inhibitory IC_50_ of α-amylase and α-glucosidase.

	Process Method	Content of Polyphenol	Content of Flavonoid	Content of Saponin	IC _50 α-amylase_ Value of Extraction	IC _50α-glucosidase_ Value of Extraction	IC_50 α-amylase_ Value of Digestive	IC _50α-glucosidase_ Value of Digestive	Content of Polyphenol of Digestive	Content of Flavonoid of Digestive	Content of Saponin of Digestive
**Process method**	1	0.477 *	0.474 *	0.425	−0.596 **	−0.557 *	−0.656 **	−0.654 **	0.660 **	0.684 *	0.489 *
**Content of polyphenol**	0.477 *	1	−0.083	−0.198	−0.894 **	−0.950 **	0.005	0.009	0.015	0.131	−0.16
**Content of flavonoid**	0.474 *	−0.083	1	0.237	−0.09	0.075	−0.578 *	−0.580 *	0.539 *	0.256	0.302
**Content of saponin**	0.425	−0.198	0.237	1	−0.039	0.195	−0.357	−0.356	0.369	0.684 **	0.993 **
**IC _50 α-amylase_ value of extraction**	−0.596 **	−0.894 **	−0.090	−0.039	1	0.936 **	0.162	0.158	−0.181	−0.342	−0.093
**IC _50α-glucosidase_ value of extraction**	−0.557 *	−0.950 **	0.075	0.195	0.936 **	1	0.041	0.038	−0.063	−0.178	0.146
**IC_50 α-amylase_ value of digestive**	−0.656 **	0.005	−0.578 *	−0.357	0.162	0.041	1	1.000 **	−0.998 **	−0.817 **	−0.418
**IC _50α-glucosidase_ value of digestive**	−0.654 **	0.009	−0.580 *	−0.356	0.158	0.038	1.000 **	1	−0.997 **	−0.815 **	−0.416
**Content of polyphenol of digestive**	0.660 *	0.015	0.539 *	0.369	−0.181	−0.063	−0.998 **	−0.997 **	1	0.845 **	0.428
**Content of flavonoid of digestive**	0.684 **	0.131	0.256	0.684 **	−0.342	−0.178	−0.817 **	−0.815 **	0.845 **	1	0.723 **
**Content of saponin of digestive**	0.489 *	−0.160	0.302	0.993 *	−0.093	0.146	−0.418	−0.416	0.428	0.723 **	1

* Significant correlation (*p* < 0.05), the Tukey-*b* test; ** extremely significant correlation (*p* < 0.01), the Tukey-*b* test.

## Data Availability

The original contributions presented in the study are included in the article, further inquiries can be directed to the corresponding author.

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
