# Peer review of "The Heating Under Micro Variable Pressure (HUMVP) Process to Decrease the Level of Saponin in Quinoa: Evidence of the Antioxidation and the Inhibitory Activity of α-Amylase and α-Glucosidase"

_foods, 2024, doi:10.3390/foods13223602_

Round 1
Reviewer 1 Report
Comments and Suggestions for Authors
The subject of the paper exploring the development of a process to decrease the level of saponins in quinoa while conserving the antioxidant properties of this potential source of proteins is currently of important interest. Indeed, saponins may be considered as antinutritional factors and the aim is to limit their content in these alternative proteins plant-based sources.
The equipment and the approach used by HUMVP to decrease the saponin content in quinoa is interesting because innovative, offering the potential to remove saponins without altering the content of beneficial compounds such as flavonoids and any other polyphenols.
The methods applied for the quantification of polyphenols, flavonoids and saponins are very well-known methods and often used in the field of polyphenols for their quantification. However, they are neither specific nor accurate as based on chemical reactions, and even though used for a category of compounds, they can measure other compounds. This is the case of the Total Polyphenols Content which can also cross react with many molecules having an antioxidant capacity. So, they are interesting methods to use to have an idea of the content in the case of polyphenols or flavonoids which are minor compounds of interest for this paper. However, it would have been more appropriate to determine the content in saponins by a more specific and precise method such as HPLC-UV to better conclude on the impact of the HUMVP on the saponin removal. It wouldn’t be expected to measure by HPLC-UV all the saponins of quinoa but at least the 2-3 major ones or the ones for which chemical standard are commercially available to allow their proper quantification.
The material and methods part of the manuscript is, to my opinion, missing an important amount of information for the reader to understand the methodologies used and interpret the data. Most of the methods used are not described in this manuscript but referred to previous papers. A recommendation would be that the authors specify if these previously described methods were applied exactly as described in the references cited with no modification at. Any modification, even if small, such as a different sample weight, should be mentioned. And even if the method used is the same as the one cited in the reference, a brief description is always useful to help understanding by the reader.
In the introduction, it is mentioned by the authors that previous methods used for removal of saponins have also investigated the impact on protein, fat, starch and dietary fibers content. The authors also mentioned on line 60 that it is of importance to find a method removing saponins “while preserving its valuable nutrients, antioxidant and functional components”. It would have been interesting that the authors investigate also the impact of their new method on nutritional values (proteins, fat, sugars) to enable the comparison with previous technics.
I would recommend that the authors introduce the reason for the choice of the two target enzymes a-amylase and a-glycosidase tested. This must be mentioned in the introduction. Why these enzymes were chosen and not other targets?
To conclude, the main suggestion would be that the authors improve the experimental part to provide all the necessary information to the reader to understand the work done and the data obtained.
Different other comments:
- All the references are duplicated between numbers and name/date along the manuscript.
- Figure 1: the numbers used to indicate what are the different elements of the equipment should be made bigger and more visible (black instead of grey).
- L133: the substrate is prepared at different concentrations: in which solvent is it prepared?
- L138: the glucose oxidase method is mentioned for the quantification of glucose released but the method is neither described in the present manuscript nor referred to a previous paper. Could the authors give more details in the material and methods section?
- Table 1: polyphenol content is reported in mg/g in the Table. Is it correct? According to Fig 2 it is in mg/100g. Could the authors check?
- L176 to L179: the quinoa treated with HUMVP is compared to Q2MM and to raw quinoa regarding the content in polyphenols, flavonoids and saponins. Could the authors explain in which figure or table are shown the values of polyphenols, flavonoids and saponins for raw quinoa and Q2MM used to calculate these retained percentages?
- L182: the amount of flavonoid should be 1.33 according to Table 1, for TPC of 327.22 mgGAE/100g and saponin content of 10.65 mg/g.
- L191-192: the optimized conditions, from what it is understood from L183-188 are obtain at run 8 (Table 1) so OH, DPPH and ABTS at these conditions should be respectively 69.89%, 91.87% and 52.18% and not 70.02%, 87.13% and 50.95% which are obtained with condition of run 11. Could the authors check if this is correct?
- Table 2: What is the concentration of the acarbose used? How was it prepared?
- Table 2: what is “digestive of quinoa samples”? What are these samples? Their identity and the way they are obtained is not clear.
Author Response
Dear reviewers
Thankas a lot for your hardwork to review oue manuscript. we have answered the comments one by one in following, please check in the attachment.
Best wish to all!
Wang anna
OCT. in 2024,
comment1: All the references are duplicated between numbers and name/date along the manuscript.
answer to comment1: We have cancelled the renferences name/date along the manuscript and keeped the reference number in red.
comment 2:Figure 1: the numbers used to indicate what are the different elements of the equipment should be made bigger and more visible (black instead of grey)
answer to comment2: we have enlarged the number in black.
comment3:L133: the substrate is prepared at different concentrations: in which solvent is it prepared?
answer to comment3: the way to prepared the differnt concentration in 2.5.1 in red letter.
comment4: Table 1: polyphenol content is reported in mg/g in the Table. Is it correct? According to Fig 2 it is in mg/100g. Could the authors check?
answer to comment 4:the polyphenol content is mg/100g in the Table1,we have corrected in red.
comment 5:L176 to L179: the quinoa treated with HUMVP is compared to Q2MM and to raw quinoa regarding the content in polyphenols, flavonoids and saponins. Could the authors explain in which figure or table are shown the values of polyphenols, flavonoids and saponins for raw quinoa and Q2MM used to calculate these retained percentages?
answer to comment 5:we have reported the content of the content in polyphenols, flavonoids and saponins in the raw qunioa and Q2MM in 2.2.
comment 6:L191-192: the optimized conditions, from what it is understood from L183-188 are obtain at run 8 (Table 1) so OH, DPPH and ABTS at these conditions should be respectively 69.89%, 91.87% and 52.18% and not 70.02%, 87.13% and 50.95% which are obtained with condition of run 11. Could the authors check if this is correct?
answer to comment 6:Thanks a lot you are right, we hvae corrected in red.
comment 7:Table 2: What is the concentration of the acarbose used? How was it prepared?
answer to comment 7:the way to prepared the differnt concentration of acarbose in 2.5.1 in red letter.
comment 8:Table 2: what is “digestive of quinoa samples”? What are these samples? Their identity and the way they are obtained is not clear.
answer to comment 8:we have reported the prepared procedure of digestve of quinoa samples in 2.2 in red.
Reviewer 2 Report
Comments and Suggestions for Authors
Authors are encouraged to specify the characteristics of the quinoa used in the study in the materials and methods. Was any characterization performed prior to the study?
I believe that most of the methodology should be described in more detail. In most cases, reference should simply be made to the methods used as reference.
Correct the methodology by using the Origin or Original program.
Table 1 describes a series of experimental treatments. Was any experimental design used? What is it? Describe whether it is a response surface, etc.
Review the correct way to write scientific names throughout the document and in the references section.
Figures require better quality and format, homogenize scales in inhibition graphs, as well as Lineweaver-Burk graphs.
Author Response
Dear reviewer
Thanks a lot for your hardwork! we have corrected the manuscript one by one , please check the attachment.
Best wish!
Wanganna
QCT. 10th in 2024
comment 1: Authors are encouraged to specify the characteristics of the quinoa used in the study in the materials and methods. Was any characterization performed prior to the study?
answer to comment 1: Yes
comment 2:I believe that most of the methodology should be described in more detail. In most cases, reference should simply be made to the methods used as reference.Correct the methodology by using the Origin or Original program.
answer to comment 2: we have corrected the methodology in red.
comment 3:Table 1 describes a series of experimental treatments. Was any experimental design used? What is it? Describe whether it is a response surface, etc.
answer to comment 3:It was respoonse surface.
comment 4: Review the correct way to write scientific names throughout the document and in the references section.
answer to comment 4: We have corrected in red.
comment 5:Figures require better quality and format, homogenize scales in inhibition graphs, as well as Lineweaver-Burk graphs.
answer to comment 5:Yes
Reviewer 3 Report
Comments and Suggestions for Authors
Dear authors,
Congratulation for you manuscript.
To enhance the quality of your work, I recommend that you edit it as follows:
Line 1-5: I suggest to the authors to take into account a new title, perhaps:
The Heating Under Micro Variable Pressure (HUMVP) process to decrease the level of saponin quinoa: evidences of the antioxidation and the inhibitory activity of α-amylase and α-glucosidase
Line 12: Please enter the quinoa's scientific name: Quinoa chenopodium (in italic). Please enter the acronym for HUMPV (Heating Under Micro Variable Pressure)
Line 28: Kindly ensure that this is the appropriate format for citing sources in the main text. Reviewing the author's guidance is advisable.
Line 42: Please provide a paragraph or two describing saponins in general and the specific saponins found in quinoa.
Line 44: Please provide a paragraph outlining wet methods—of course with examples—for removing saponins.
Line 48: Please provide one or two instances.
Line 70-72: Kindly insert the following informations: Trolox (6-Hydroxy-2,5,7,8-tetramethylchromane-2-carboxylic acid), DPPH (2,2-Diphenyl-1-picrylhydrazyl), ABTS (2,2'-azino-bis(3-ethylbenzothiazoline-6-sulfonic acid)
Line 75: It's unclear how many samples—three or seventeen—you used for this investigation. Please put them all in a table and make sure each one's experimental variables are clear.
Kindly insert one more section:
2.2. Samples performed in this study
2.3. Processing of quinoa grains
Line 94-95: Figure 1: Kindly allow the pictures to expand. It is a little difficult to see their letters or numerals.
Line 96-106: Please used the recommendations to update the text.
Line 108: Could you briefly explain the technique used?
Line 111: and the results were expressed as ....
Line 117: % ? what kind of starch do you used in this study?Could you briefly explain the technique used?
Line 119: Sections 2.5.1 and 2.5.2 might consist of.
Could you briefly explain the techniques used?
Line 122: and α-glucosidase
Line 127: please insert a pause in between them\
Line 134: the
Line 136: . After that, the samples were incubated ....
Line 140: What does Km mean?
What does Vmax mean?
Line 159-161: Figure 2: Kindly allow the pictures to expand. It is a little difficult to see their letters or numerals. For example, the arrangement of the four graphs could be in two or three lines.
Line 162: Table 1: Typically, the investigated parameters' abbreviations are used in the tables after the unit of measurement to give a clearer image of the results obtained. Please used the recommendations to update the text.
Line 186: unit of measurement ?
Line 195: Would you perhaps provide one or two comparable references to back up your data?
Line 231: amylase, glucosidase
Line 232: Please remove the space that exists between the table and the title of Table 2.
Why did you choose this bold line?
Line 263: Table 3: Typically, the investigated parameters' abbreviations are used in the tables after the unit of measurement to give a clearer image of the results obtained. See the recommendations.
Line 283: Figure 3: Kindly allow the pictures to expand. It is a little difficult to see their letters or numerals. For example, the arrangement of the six graphs could be in two or three lines.
Line 294: Figure 4: Kindly allow the pictures to expand. It is a little difficult to see their letters or numerals. For example, the arrangement of the six p graphs could be in two or three lines.
Line 314: Kindly add one more conclusion:
In conclusion, this study's primary goal was to assess the impact of HUMPV conditions on quinoa's ability to retain nutrients while lowering saponins and the findings support this purpose.
Line 315: Please specify each author's contribution in this manuscript, including who made what.
https://www.mdpi.com/journal/foods/instructions
For research articles with several authors, a short paragraph specifying their individual contributions must be provided. The following statements should be used "Conceptualization, X.X. and Y.Y.; Methodology, X.X.; Software, X.X.; Validation, X.X., Y.Y. and Z.Z.; Formal Analysis, X.X.; Investigation, X.X.; Resources, X.X.; Data Curation, X.X.; Writing – Original Draft Preparation, X.X.; Writing – Review & Editing, X.X.; Visualization, X.X.; Supervision, X.X.; Project Administration, X.X.; Funding Acquisition, Y.Y.”,
Line 324: Please remove the underlined phrase if you do not have a Data Availability Statement for this publication.
Line 326: That is not, in my opinion, the proper way to write the references. Kindly carefully review the author's guide.
https://www.mdpi.com/journal/foods/instructions
e.g. Journal Articles:
1. Author 1, A.B.; Author 2, C.D. Title of the article. Abbreviated Journal Name Year, Volume, page range.

Author Response
Dear reviewer
Thanks a lot for your hardworking. We have revised our manuscript according to your comments.
please check it in the attachment.
Best wish to you all!
Wanganna 12 Qct. in 2024.
comment 1: Line 1-5: I suggest to the authors to take into account a new title, perhaps:
The Heating Under Micro Variable Pressure (HUMVP) process to decrease the level of saponin quinoa: evidences of the antioxidation and the inhibitory activity of α-amylase and α-glucosidase
answer to comment 1: we have change the title according to your opinions.
comment 2:Line 12: Please enter the quinoa's scientific name: Quinoa chenopodium (in italic). Please enter the acronym for HUMPV (Heating Under Micro Variable Pressure)
Line 28: Kindly ensure that this is the appropriate format for citing sources in the main text. Reviewing the author's guidance is advisable
answer to comment 2: we have done
comment 3:
Line 42: Please provide a paragraph or two describing saponins in general and the specific saponins found in quinoa.Line 44: Please provide a paragraph outlining wet methods—of course with examples—for removing saponins.Line 48: Please provide one or two instances.
answer to comment 3: the function of the compounds not the single of phenolics, flavonoid, saponin in food should be taken into account, so we just focus on the compounds in this paper.
wet methods to remove the saponin is washing the quinoa grain in clear water, a simple way like washing rice before cooking.
comment 4:
Line 70-72: Kindly insert the following informations: Trolox (6-Hydroxy-2,5,7,8-tetramethylchromane-2-carboxylic acid), DPPH (2,2-Diphenyl-1-picrylhydrazyl), ABTS (2,2'-azino-bis(3-ethylbenzothiazoline-6-sulfonic acid)Line 75: It's unclear how many samples—three or seventeen—you used for this investigation. Please put them all in a table and make sure each one's experimental variables are clear.
answer to comment 4:we have finished that according to your comment.
comment 5: 2.2. Samples performed in this study/2.3. Processing of quinoa grains
answer to comment 5: we have revised the sample and processing of quinoa grain in red letters.
comment 6:Line 159-161: Figure 2: Kindly allow the pictures to expand. It is a little difficult to see their letters or numerals. For example, the arrangement of the four graphs could be in two or three lines.
Line 283: Figure 3: Kindly allow the pictures to expand. It is a little difficult to see their letters or numerals. For example, the arrangement of the six graphs could be in two or three lines.
Line 294: Figure 4: Kindly allow the pictures to expand. It is a little difficult to see their letters or numerals. For example, the arrangement of the six p graphs could be in two or three lines.
answer to comment 6:we have expand the figure 2\3\4 to allow they clear.
comment 7:Line 314: Kindly add one more conclusion:
In conclusion, this study's primary goal was to assess the impact of HUMPV conditions on quinoa's ability to retain nutrients while lowering saponins and the findings support this purpose.
answer to comment 7:we have finished it .
comment 8:Line 315: Please specify each author's contribution in this manuscript, including who made what. https://www.mdpi.com/journal/foods/instructions
For research articles with several authors, a short paragraph specifying their individual contributions must be provided. The following statements should be used "Conceptualization, X.X. and Y.Y.; Methodology, X.X.; Software, X.X.; Validation, X.X., Y.Y. and Z.Z.; Formal Analysis, X.X.; Investigation, X.X.; Resources, X.X.; Data Curation, X.X.; Writing – Original Draft Preparation, X.X.; Writing – Review & Editing, X.X.; Visualization, X.X.; Supervision, X.X.; Project Administration, X.X.; Funding Acquisition, Y.Y.”,
answer to comment 8: we have done.
comment 9:Line 324: Please remove the underlined phrase if you do not have a Data Availability Statement for this publication.
Line 326: That is not, in my opinion, the proper way to write the references. Kindly carefully review the author's guide.https://www.mdpi.com/journal/foods/instructions
answer to comment 9:we have done.
Round 2
Reviewer 1 Report
Comments and Suggestions for Authors
Thanks to the authors for having answered all the comments and questions.
Author Response
dear reviewer
Thanks a lot for your hard work, we have revised the manuscript in red.
Best wish
Wangann, 16 Oct. in 2024
Reviewer 3 Report
Comments and Suggestions for Authors
Dear Authors,
Please find the revised manuscript.
To enhance the quality of your work, I recommend that you edit it as follows
Line 26-34: You ought to, in my opinion, include a gap between the characteristics and [1,2]. The final word and dot are joined without a space, as follows "characteristics [1,2]." Please fill out the entire document using the correct form.
Line 56: Why here is a free line?
Line 62: alpha=α
Line 87-93, 111, 122, 268: Please insert/ remove the space between them. See the manuscript.
Line 188: Table1 - Please enclose all measurement units in round brackets.
Line 255: Table 2 - Why is Table 2 in this location? I would like it to be placed on page 9.
Line 306- Table 3, Line 319- Table 4: Please arrange them in three lines. Reading them will be simpler.
Line 353: Please remove the underlined phrase if you do not have a Data Availability Statement for this publication.
My resolution for this manuscript is: Minor revision.
Best regards,
Reviewer

Author Response
Dear reviewer
we have correct manuscript according to "the comments: To enhance the quality of your work, I recommend that you edit it as follows".
please check it again.
Best wish
Wangann , 16 Oct. in 2024